# An Informed Decision Support Framework from a Strategic Perspective in the Health Sector

Mohammed Alojail [1], Mohanad Alturki [1] and Surbhi Bhatia Khan [2,*]

1   Department of Management Information Systems, College of Business, King Saud University, P.O. Box 28095, Riyadh 11437, Saudi Arabia; malojail@ksu.edu.sa (M.A.); mohanad.alturki@gmail.com (M.A.)
2   Department of Data Science, School of Science, Engineering and Environment, University of Salford, Salford M5 4WT, UK
*   Correspondence: surbhibhatia1988@yahoo.com

**Abstract:** This paper introduces an informed decision support framework (IDSF) from a strategic perspective in the health sector, focusing on Saudi Arabia. The study addresses the existing challenges and gaps in decision-making processes within Saudi organizations, highlighting the need for proper systems and identifying the loopholes that hinder informed decision making. The research aims to answer two key research questions: (1) how do decision makers ensure the accuracy of their decisions? and (2) what is the proper process to govern and control decision outcomes? To achieve these objectives, the research adopts a qualitative research approach, including an intensive literature review and interviews with decision makers in the Saudi health sector. The proposed IDSF fills the gap in the existing literature by providing a comprehensive and adaptable framework for decision making in Saudi organizations. The framework encompasses structured, semi-structured, and unstructured decisions, ensuring a thorough approach to informed decision making. It emphasizes the importance of integrating non-digital sources of information into the decision-making process, as well as considering factors that impact decision quality and accuracy. The study's methodology involves data collection through interviews with decision makers, as well as the use of visualization tools to present and evaluate the results. The analysis of the collected data highlights the deficiencies in current decision-making practices and supports the development of the IDSF. The research findings demonstrate that the proposed framework outperforms existing approaches, offering improved accuracy and efficiency in decision making. Overall, this research paper contributes to the state of the art by introducing a novel IDSF specifically designed for the Saudi health sector.

**Keywords:** decision support; framework; stakeholders; information systems; analysis

## 1. Introduction

The decision-making process is crucial for the success and efficiency of organizations, requiring effective management, analysis, and communication to generate optimal solutions [1]. Poor decisions can have detrimental effects on an organization's performance, leading to wasted time, resources, and financial loss. To aid managers in making informed decisions and maintaining control over strategic initiatives, decision support frameworks and systems (DSS) have been developed [2]. This research focuses on proposing an informed decision support framework (IDSF) specifically designed for Saudi organizations, aiming to enhance their decision-making processes and contribute to the development of comprehensive decision support systems. In this context, the objectives of this study are twofold. Firstly, it aims to review the concept, background, and history of effective decision making and DSS, while also considering relevant studies in the field. By doing so, it seeks to identify gaps in the existing literature that this paper intends to fill. Secondly, the research involves conducting interviews with stakeholders within Saudi organizations to gain insights into their perspectives on decision making processes, as well as assessing

their current needs and expectations regarding decision support frameworks. The analysis of the interview results will inform the development of an IDSF tailored specifically for Saudi Arabian organizations, addressing strategic decision-making processes. Additionally, this study aims to establish a comprehensive implementation process to facilitate consistent and effective decision making throughout all stages of an organization's development.

Effective decision making is critical for individuals and organizations alike, as it directly impacts their overall performance. Therefore, there is a need for an informed decision support framework that can leverage data-driven insights and consider both internal and external factors of the business [3]. Political and economic influences, customer preferences, and the integration of digital and non-digital sources are among the crucial aspects that organizations must consider adopting an integrated approach to decision making. By leveraging technological advances, such frameworks guide and facilitate decision-making processes, optimizing outcomes through accurate data collection and sophisticated analytical capabilities [4]. Furthermore, recent years have witnessed a significant focus on methods and techniques that promote decision making in organizational management. Decision support systems (DSS) have evolved from individual user tools to shared resources across organizations, leveraging computer systems and the internet. These computer-based systems aim to improve productivity and efficiency, supporting decision-makers and policy developers in long-term planning while providing flexibility in interactions with multiple users [5].

In recent years, Saudi Arabia has experienced rapid economic and technological progress, positioning its organizations as global competitors. Notably, Saudi Aramco became the world's largest and most valuable company by market capitalization in May 2022 [6]. In light of these developments, Saudi organizations must adapt their decision making processes to effectively handle the increasing influx of data involved. Compiling and analyzing such data can be challenging for top leadership teams, emphasizing the need to develop systems and frameworks that promote decision making effectiveness and efficiency [7]. Therefore, organizations in Saudi Arabia are increasingly focusing on improving their decision-making processes to remain competitive in the rapidly changing global economy. Thus, this paper addresses the need for an informed decision support framework tailored for Saudi organizations. By proposing an IDSF specifically designed for the Saudi Arabian context, this study aims to fill gaps in the existing literature and provide practical implications for decision makers. The following sections present a comprehensive analysis of the state of the art, research design, data collection methods, evaluation metrics, and results, demonstrating the originality and superiority of the proposed framework over existing approaches.

## 1.1. Motivations and Contributions

The aim of this research is to develop an informed decision support framework for the Saudi Arabian organizations, taking the case of health sector. In addition, the research should answer two fundamental questions which are: how the decision makers ensure the accuracy of the decision that has been chosen? what is the proper process that assure the accuracy of the decision? However, the objectives of this research are the following:

- To review the decision support system framework history, concept, and case studies in the related area;
- To conduct interviews with relevant stakeholders;
- To analyze the interviewees' responses to the interview;
- To develop an informed decision support framework to oversee and tackle strategic decisions for the decision maker in a Saudi organization;
- To visualize the findings and trends using a flow diagram for the decision making process.

## 1.2. Paper Organization

This paper is divided into six sections to provide a comprehensive understanding of the proposed decision support framework for Saudi organizations. The first section, the

Introduction, sets the context of the analysis, highlights the research focus, specifies the scope of the paper, and identifies the gap in the literature that the study aims to address. The second section, Related Works, reviews the existing literature on decision making processes, decision support systems, and relevant studies in the area. The third section, Proposed Framework, presents the conceptual framework designed to enhance decision-making processes in Saudi organizations. The fourth section, Methodology, outlines the research approach, including data collection methods and analysis techniques. The fifth section, Findings, presents the results of the study, including insights obtained from interviews and the analysis of data. Finally, the sixth section, Discussion and Conclusion, discusses the implications of the findings, highlights the novelty of the results, and concludes the paper by recommending the implementation and adaptation of the proposed framework.

## 2. Related Work

The history around the decision support framework as well as the DSS framework is discussed, and so are the different cases and examples discussed in previous research mainly focusing in the countries of the Gulf including Saudi Arabia.

Decision support systems (DSSs) or frameworks have evolved over the years from simple model-oriented systems to the current advanced multi-function entities. In the early days, in the 1960s, it was expensive to construct a large-scale information system. However, new systems such as the IBM System 360 came up and were powered by powerful mainframe systems [8]. It became practical and feasible to develop management information systems (MIS) for large organizations. These systems focused on offering the structured management reports, periodically [8]. At the time, the main focus was on the accounting and transaction systems. As technologies developed, additional systems developed with a specific focus on assisting management in decision making processes. The management decision system (MDS) was developed between 1966 and 1967 by Scott Morton, and it used computers and analytical models [8]. Later in 1971, Scott Morton and Gory Keen coined the term decision support system (DSS) as cited in their 1978 book, "Decision support systems: An organizational perspective" [9]. As such, DSS refers to an interactive software system, which provides information derived from models and data in a way that enables decision makers to solve decision problems more effectively [9]. Consequently, it is a framework or a system, which assists the decision maker, but it does not replace them. The applications of DSS are extensive and they include both structured, semi-structured, and unstructured processes. The development and utilization of DSS frameworks was to enable decision makers consider more aspects and options in the decision making process and prevent tunnel vision [9].

While the early research in the field contributed massively to the current systems and operations, the last few decades have led to huge developments in the field. Since the 2000s, there have been various major changes in the DSS theory and practice. Some of the developments include the incorporation of business intelligence (BI) and business analytics (BA) concepts in DSS [10]. The concepts focus on different aspects of using computer systems to gain information and analyze it. As computers became increasingly powerful, companies integrated them into their daily operations, and they could capture massive amount of data, which could be analyzed further to guide decision makers in their endeavors.

The DSS industry has transformed over the years resulting in the development of an advanced decision theory. In 2002, Daniel Kahneman received the Nobel Prize for the decision making theory, which he developed together with Amos Tversky. The theory is based on a set of theories that explain the cognitive processes of how humans make decisions, with a specific focus on system failures [10]. It was one of the main transformations that resulted in DSS taking a scholarly and research perspective. Theoretical contributions in the field have led to the collection of data and continued analysis to ensure the information gathered answers numerous questions and allows further developments within the field.

*2.1. Decision Support Frameworks Concept*

The decision making process requires the integration of multiple conflicting and non-measurable dimensions. As a result, one of the emerging and commonly adopted aspect of DSS is multi-criteria decision analysis (MCDA), and it addresses an alternative to dealing with complex decision making issues, which include multiple, diverse, and conflicting goals [11]. There are DSS tools developed to support the approaches used in MCDA to facilitate decision making processes using data via models for the resolution of semi-structured and unstructured problems. The tools allow a decision maker to map out all possible alternatives to a decision. To allow such analysis, computer-based modeling has been the main area of focus in DSS research [11]. As discussed in the background section, it has been in place and followed the development of computers. Since the mid-1970s, computer-based modeling started appearing, and it used web technologies and modeling software [12]. In the beginning, they were not as sophisticated as the tools available today [11]. The applications of the computer-based models have been used in multiple industries including agriculture, climate change, food, medicine, and supply chain, among others. Cloud storage and access to information through multiple devices globally made web-based technologies emerge as the newest trends within the computer-based modeling arena [11]. DSS requires the use of computerized information systems, which include expert systems (ES) and MCDA, and their role is to support decision makers to use data, models, and technologies during their decision making processes. As such, it has led to the use of data-driven DSS, which is mainly focused on data interpretation. Expert systems (ES) are also referred to as knowledge-based systems (KBSs) and are rule-based software programs focusing on a specific problem domain. They have incorporated the use of the web where databases are used in the storage and processing of data. When a user accesses data through a web-based DSS, access is granted through a central server system, and in recent years, this has been carried out through web browsers. The integration of previously complex IT concepts into user-friendly models has made it easier to incorporate DSS into the workplace [11].

Based on these concepts of the DSS, it is evident that the different generations have emerged due to demand from organizations and departments to have the right tools and techniques to support complex decision making processes. In most cases, such decisions are marred by risks and uncertainties, requiring the integration of human intelligence, IT, and software to interact with each other for the overall benefits of the entity. These DSS frameworks are distinguished from other IT systems through their integration of technology and operations research within a decision makers' competence structure [13]. Furthermore, this leads to an increasing number of alternatives and the possibilities of selecting the optimal alternatives from a set of tested options, which offers rapid sensitivity analysis and response. Since these frameworks involve the incorporation of computer systems, they can provide support for successive and interconnected decision series. Therefore, throughout all the decision making stages, sufficient support is accorded to the decision maker. This also improves the overall business understanding of the decision makers, which also involves visualizing relationships and thus visualizing a comprehensive business image. Business operatives can also give rapid responses to unexpected situations as they can access forms and variables with ease. Business managers are also prepared with the capacity to perform necessary analysis for a particular purpose, which also provides them with a variety of technical means and approaches for the preparation of analysis for specific business needs [13]. This prepares businesses with improved communication and oversight capacities, and the communication channels are also well-documented, which concludes in increased consistency of planning and standardized accounting procedures [13]. This also means that companies can make better decisions, improve teamwork, and use available data efficiently. Finally, it also saves time and costs as decisions made using these models are thought to be highly reliable. Therefore, an organization can have an advantage over its competition by incorporating DSS in its processes.

Understanding these concepts also requires an appreciation of the aims and principles that most of these DSS frameworks possess. Since DSS enables problem resolutions to various problems, and quick responses to unexpected situations, organizations manage to operate efficiently in dynamic work environments. These decision support capabilities allow the resolution of unstructured and semi-structured issues, which improves the management's expertise and knowledge on matters. Therefore, as illustrated by multiple researchers, DSS should be used in organizations' decision support systems to promote assistance to the management in dealing with complex and semi-structured problems. It should also help them in making decisions rather than altering them. finally, it should be a source of effectiveness and efficiency during the decision making process.

### 2.2. Benefits and the Need of Decision Support Framework in Saudi Arabia

Increasing globalization and the technological advancements of the 21st century have caused businesses in Saudi Arabia to evolve in order to remain competitive. As such, decision support frameworks have become increasingly relevant in helping organizations and decision makers to gain insight into their decision making process and optimize operations [2]. An information system designed to support decision making by collecting, organizing, and presenting data in a meaningful way provides decision makers with a clear view of the decision making process, enabling them to make more informed decisions quickly and accurately. It can help organizations identify areas of potential improvement by providing key insights into costs and performance through interactive data visualization tools.

Throughout the literature review process and analysis of the conceptual framework, the literature here identified two main gaps, which motivate the current research, and the eventual development of an ideal decision support framework for use in Saudi Arabian organizations.

### 2.3. Factors That Affect the Development of Decision Support Framework

There are several factors that can impact the development of a proper decision support framework. These factors can include the culture and values of the organization, how hard it is to make decisions, the quality and availability of data, and the skills and knowledge of the team that is in charge of putting the framework into place among others.

One of the major elements that can affect the development of a proper DSS framework is the team skills and knowledge. Setting up a new system requires the team to have the necessary skills and knowledge to understand its importance and how to use it [14–18]. Decision making in an organization is a team job as the person making the final decision requires the input of others within the team and at different levels within the organization. Consequently, it is critical for the organization to have individuals with the necessary team skills to operate the framework efficiently. In addition to the skills needed, the technical requirements of the project also affect its success. An extensively technical system may be challenging to implement and might also require intensive training for the users [19,20].

A competent framework's ability to be developed might also be impacted by other significant factors, such as the quality and availability of data. DSSs rely on data to make decisions. Therefore, the effectiveness of the system can be significantly impacted by the correctness and dependability of the data used by DSS [11,15,21,22]. Organizations need to ensure that they have access to high-quality data that are relevant to the decisions being made and that they have systems in place to regularly review and update the data used by DSS [22]. Therefore, successful implementation also requires the organization to invest in data collection and management systems, which guarantee quality. Another critical factor is the complexity of the decision making process which can impact the development of a proper support framework. According to the authors of [23], the DSS works best when used to support relatively simple and structured decision making processes. When the DSS framework developed is complex with multiple connecting points, it might be impossible to operate for employees and potential users [16,20,23]. When designing a framework, it is

critical that it be made as simple as possible, and also involve the users during the process to ensure they understand how it will integrate into their way of operating. The framework should be made in a way that improves service delivery and decision making and does not make operations challenging for the users [23]. Some of the researchers defined the method and demonstrated the interconnections between open innovation and intellectual property [24].

Another element relates to the cost–benefit analysis of the decision made. The financial element of such a project according to the decision made determines its potential to implement and be sustained in an organization. When developed for a specific organization, the system needs to have positive cost–benefit analysis [9,19,20]. In addition to all the factors, leadership support is also required [13,16]. Most DSS frameworks are meant for use by leaders to inform their decision making processes. Therefore, when leaders support the implementation, there is potential for the successful implementation of the project.

According to the authors of [14], organizations with a strong culture of data-driven decision making may be more likely to adopt and effectively implement DSS [14]. On the other hand, organizations with a more traditional or reactive approach to decision making may be less receptive to have the decision support framework or may struggle to effectively integrate the system into their decision making processes. It is important for organizations to consider their culture and values when developing the decision support framework and to ensure that the framework is aligned with these values and is able to effectively support the organization's decision making processes [14].

Finally, the time is an essential factor in the framework's implementation [16]. The framework or the system require to be implemented in a timely nature to allow the analysis of the data and allow decision makers to obtain solutions. In addition, when the system saves time, it is easily integrated into the organization [16]. Ultimately, these factors are critical in the ultimate implementation and it may not be easy to work with them in isolation. A system must support or align to most, if not all these factors. The summary of the factors is given in Table 1.

**Table 1.** Summary of the factors that affect the development of decision support framework.

| Factor Name | Definition | Source |
|---|---|---|
| Team skills and knowledge | Skills and knowledge developed towards the use of the new framework. | [14–18] |
| Quality of Data Available | The value added via the available data and their reliability | [11,15,21,22] |
| Complexity | The complication of the decision making process | [16,20,23] |
| Cost–benefit analysis | The decision made needs to compare with the benefits it gives | [9,19,20] |
| Leadership Support | Leadership understanding and support of the whole process | [13,16] |
| Communication | Inter-relationship among the parties involved | [16,24,25] |
| Technical requirements | Systems and installations needed for the system to operate | [19,20] |
| Organizational data-driven culture and values | The way of doing things within the organization | [14] |
| Time | Adequate factor analysis time | [16] |

Identifying the factors, the investigations were also conducted on determining what methods are effective and could be used in the analysis. The summary of the source is presented in Table 2.

**Table 2.** Techniques used for performing the investigations.

| Source | Techniques Used | Pros | Cons |
|---|---|---|---|
| Zong et al. (2021) [1] | Experiments | Improves decision making in e-commerce | Limited application domain, may not generalize well |
| Aversa et al. (2018) [2] | Case Study | Provides insights into strategic information systems | Specific to the context of Formula 1 |
| Gupta et al. (2022) [3] | Longitudinal Study | Highlights AI's potential in decision support | Lacks specific dataset/application |
| Martins et al. (2019) [4] | Longitudinal Study | Facilitates decision-making in business competitions | Limited to the context of business idea competition |
| Allaoui et al. (2019) [5] | Surveys | Supports collaboration planning | Focuses on sustainable supply chains, not general DSS use |

### 2.4. Decision Support Framework Case Studies

Informed decision support frameworks are designed to assist individuals or organizations in making informed decisions by providing relevant information, analysis, and recommendations. These frameworks can be applied in a wide range of contexts, including healthcare, finance, education and technology. Below are several case studies of informed decision support frameworks from around the world. Understanding these applications is critical in identifying the gap, which is needed for further research in Saudi Arabia.

The United Kingdom Environmental Observation Funder (UK-EOF) is an organization that has set up its decision support framework with the intent of improving the decision making processes with the public sector, and especially in environmental management.

The decision support framework for the UK-EOF shown in Figure 1 is a process that captures and summarizes the key evidence needed to make a decision based on a set of common criteria or issues. The process involves input from a variety of organizations and is coordinated by a central support body, such as the UK-EOF secretariat. The process has six stages, including proposal initiation, evidence gathering, discussion forum, outputs formulated, the decision made, and observation activity catalogue modification [26].

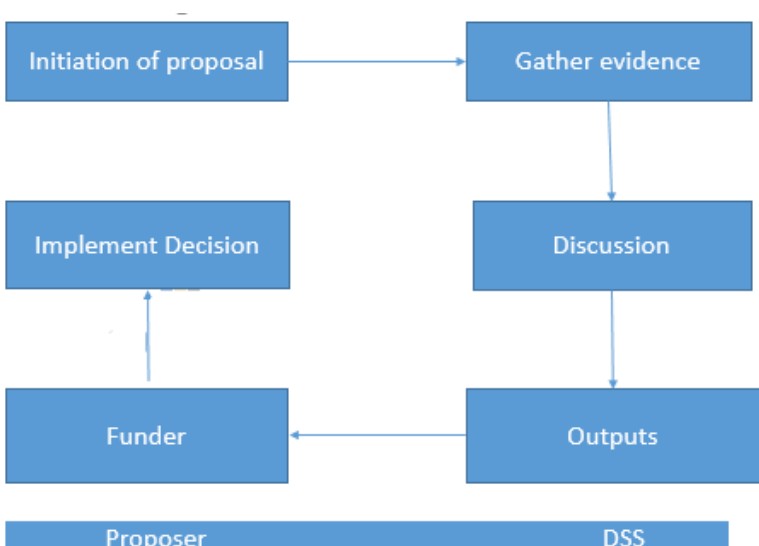

**Figure 1.** UK-EOF decision support framework source: environment research funders' forum, UK, -environmental observation framework.

The framework begins with proposal initiation, where the proposing organization or funder assigns a member of the organization to lead the proposal. The next stage is evidence gathering, where the proposing organization or funder provides the required supporting

activity description and completed draft activity scorecard, which is then circulated to external organizations for comment and scoring. The central support body then gathers the evidence, sets up a discussion forum, and provides a summary of informed responses for the proposing organization to use in making a decision. The discussion forum is the next stage, where a summary scorecard is produced using the evidence gathered. The outputs are then formulated, and the proposing organization or funder makes a decision based on the informed advice provided by the central support body and supporting organizations. The final stage is the observation activity catalogue modification, where the funder is encouraged to make any necessary changes [26].

Marušák et al. [19] discussed a DSS approach implemented in the Czech Republic. The system was named optimal, and its logical structure was based on the need for its application, which was within the forest management sector. It involved making the necessary data inputs, which included the forest stand map and forest inventory data at the start before adding information on the systems requirements, and it gave the potential harvest units [19]. It was noted as a powerful system for harvest scheduling. The authors also added that the system allowed forest managers to change the parameters and create various scenarios within a matter of minutes to find the best solutions based on specific needs [19].

Another case of the informed decision framework, it is for improving evidence-informed decision-making (EIDM) in health service management is a comprehensive framework that takes into account all the relevant factors that influence the practice of EIDM in different types of organizations [27]. This framework is designed to provide guidance on the strategies that need to be developed and evaluated to improve EIDM in health service management. The framework is based on an understanding of the various factors that interact to influence EIDM and the relationships between these factors [27].

The framework takes into consideration all factors relevant to the various types of organizations that play a significant role in influencing EIDM. These organizations include government departments, healthcare organizations, professional and training organizations, and university and research institutions. Within each type of organization, there are various factors that affect the practice of EIDM, but it is clear that the factors relevant to each type of organization are interrelated. Therefore, to best influence the practice of EIDM amongst health services managers, changes should be introduced within the three types of organizations as detailed in the framework [27].

The framework suggests that changes should be specific and relevant to the local context, making evidence more easily understood and interpreted by managers for immediate use. This focus on promoting and rewarding the use of evidence, as well as improving the relevance of evidence, ensures that managers are making informed decisions based on the most current and relevant information available. Additionally, the framework takes into account the interrelated nature of factors relevant to each type of organization, highlighting the need for a holistic approach to improving EIDM in health service management [27].

Another model framework was applied in the University of Babylon to assist in the procurement decisions highlighted in [17]. Their proposed a framework was based on data collection from the interview and consequent analysis, as well as a literature review. It is applicable in both simple and complex decision probabilities to assist in the provision of accurate results for each criteria [17].

The framework has five constructs with each of them passing through five unique stages. It starts with the initial goods evaluation in stage one followed by the development of goods evaluation in stage two [17]. Stage three requires vendor bid evaluation followed by vendor selection in stage four, and finally the supplements in stage five [17].

One of the models applied in a different setting was developed by Van Delden [9].The model incorporates three different elements, which are design, development, and implementation. (a) represents the relationship between the main parties that are involved in the development process to enable the functionality of the decision support framework. Each party has their responsibility within the process expressed. It also identifies all the commu-

nication blocks that could have the potential to prevent its development. (b) describes the development process, which is presented as an iterative process as opposed to a waterfall one. Finally, (c) places the tasks in their respective order as necessitated in the development process, and it also focuses on iterative processes that arise throughout the framework.

Di Mateo [18] proposed a decision support model for the management of cultural heritage. Their proposed framework aimed at supporting organizations and companies with interest in cultural heritage and museum management to assist in the adoption of scientific policies and criteria in their plans and management of daily operations [18].

Another decision support framework applied within the clinic sector is the multi-agent clinical decision support system which uses case-based reasoning (CBR) [28]. The clinical decision support system, CDSS, was created as an approach towards the improvement of medical decisions by focusing on clinical knowledge, patient information, and related medical information [28]. The approach integrates CBR into CDSS through a connection of the search agent to the decision agent.

The process of combining CBR into CDSS requires connecting the search agent to the decision agent. The search agent allows one to find the cases that are the most similar to the problem. An adaptation agent follows and they determine the differences between the selected cases and the current problem. If it is proven necessary, they set the necessary rules, which makes it possible to apply old solutions to the new problem [28]. An enhancement agent adapts, checks and criticizes the results and the execution agent applies the refined solution. In the end, an evaluator is responsible for storing the results in a database and the result is shared with the decision agent [28].

Finally, the application in the field of clinical medicine was explored by [20] in Saudi Arabia. They explored the application and experience of the clinical decision support system (CDSS) and its effectiveness within the healthcare sector in Saudi Arabia [20]. To be successful, the system required three main areas of focus: input content, the integrity of CDSS, and output advice. The input content had to be right, reliable, and updated. For the system to express the right level of integrity, it needed to integrate with health information system, clinical workflow, and adopt mechanisms of alerts. Finally, the output advice needed to be simple, speedy, and with references [20]. The CDSS alerts were both active and passive, and they had three main levels, which included critical, moderate, and least important. In KSA, it was implemented as part of the evidence-based medicine for the improvement of patient safety. It offers clinicians with the necessary knowledge of the specific patients or diseases which facilitate taking the decision. The summary of the DSS is shown in Table 3.

**Table 3.** Summary of the decision support framework presented.

| Framework Name | Specialty | Country | Source |
|---|---|---|---|
| Environmental observation framework (EOF) | Environmental Management in Public Sector | United Kingdom | UK-EOF [26] |
| Optimal | Forest Management | Czech Republic | Marušák et al. [19] |
| Evidence-informed decision making in health service management framework | Health Service Management | Australia | Liang et al. [27] |
| UOB DSS procurement framework | University Setting | Iraq | Abid et al. [17] |
| Decision support system development framework | Natural Hazard Mitigation | Australia | Newman et al. [9] |
| DSS framework for cultural heritage | Cultural Heritage Management | Italy | Di Mateo et al. [18] |
| Multi-agent clinical decision support system using case-based reasoning | Clinical Sector | Ukraine | Korablyova et al. [28] |
| Clinical Decision support system (CDSS) | Clinical sector | Saudi Arabia | Alqahtani et al. [20] |
| Technology acceptance model | Wearable Devices | Italy | Magni et al. [21] |

## 3. Proposed Framework

One of the main gaps indicated that most models focused on specific sectors, such as health, forestry, education, etc., for specific organizations, which reduces their application in other settings. Additionally, the literature did not find any clearly published framework that could generally be applied to the decision making processes for leaders and decision makers in Saudi organizations and serve a business decision. In addition, a comprehensive process of decision making was required to be conducted. Therefore, the research aimed and worked with the data collected to formulate and identify an ideal that would apply in Saudi organizations to foster decision making. Although there are multiple frameworks for improving the process of informing and the quality of the decision making, there is a lack of comprehensive overview of the factors that impact the decision quality and accuracy. As mentioned in the factors previously, one of the key factors are related to the quality of data availability. Therefore, to answer the research question, of how the decision makers assure decision accuracy covering all the aspects, including the non-digital sources and the data management authority which have an impact on the decision in addition to the digital sources, the proposed framework should be taken into account. Moreover, to ensure the effectiveness of the framework, a defined set of decision criteria must be applied to the decision's issue.

The proposed informed decision support framework (IDSF) for Saudi organizations, taking the health sector as an example, shown in Figure 2, outlines a comprehensive process for tackling strategic decisions at the operational, managerial, and strategic levels. The IDSF, comprising an informed decision support (IDS) committee, will be responsible for overseeing and implementing the framework. IDS will oversee and tackle strategic decisions and align with the organization's leadership, graduating tracking sub-entities' strategic decisions. IDS will act as a strategic informed decision support unit, capitalizing on the existing capabilities and resources to lead decision makers towards the best possible decisions

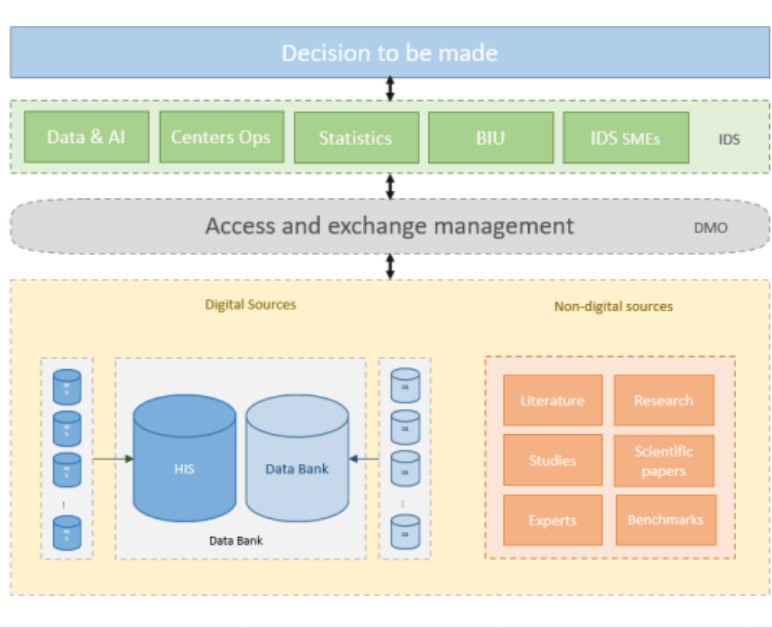

**Figure 2.** Proposed informed decision support framework (IDSF).

A notable aspect of the framework is the ability to focus on structured, semi-structured, and unstructured decisions. Structured decisions are those that are repetitive and routine, and for which a definite procedure can be followed. Semi-structured decisions involve a mix of clear-cut answers provided by accepted procedures and the need for judgment,

evaluation, and insights. Unstructured decisions are those that require the decision maker to provide judgment, evaluation, and insights into the problem definition. By addressing all three types of decisions, the framework aims to provide a comprehensive approach to informed decision making.

The framework also outlines a set of criteria for selecting decisions for consideration in order to ensure decision accuracy and efficiency, including the need for supporting data and resources, the frequency of the decision, and the minimal impact of external factors. This helps to ensure that the IDS committee is able to focus on those decisions that are most suitable for informed decision making.

There are several components of the framework, which include data and AI, which refer to the data and AI responsible for data storage and retrieval. The second component is the center's operation, which is responsible for the monitoring all the health centers and the services provide and their performance. The business intelligence unit (BIU) is another component, which is specific for the health cases and events. It has vital information and reveal information related to the statistics and IDS SME. As represented in the figures above, the criteria for the framework involve deterministic problems and no open-ended problems. Therefore, this means that there are no decisions to resolve general issues. It was critical to ensure that the framework could access the necessary supporting data and resources. Furthermore, any periodic or frequent one-time decisions must also be avoided. The external factors should also be minimal, focus on issues and have the least legal commitments or involvement.

The process flow of the framework shown in Figure 3 is thorough, starting with the receipt of a problem or question from the leadership and proceeding through stages including problem definition, data requisition and approval, data acquisition, scenario/research implementation, and dissemination. Each stage is subject to review and approval by the IDS committee and relevant execution teams, ensuring that the process is well-coordinated and that decisions are thoroughly researched and evaluated, which answers the research question of how to assure decision accuracy through a proper process. The work involves the formulation of the problem, checks its feasibility to further report it as a problem statement, and finally carries out further checks on defining it and proposing the relevant solution for the same problems. After the solution is approved from IDS, the next step is to collect the data and information needed to formulate the research questions so that it can be implemented and approved for dissemination and for making further actions.

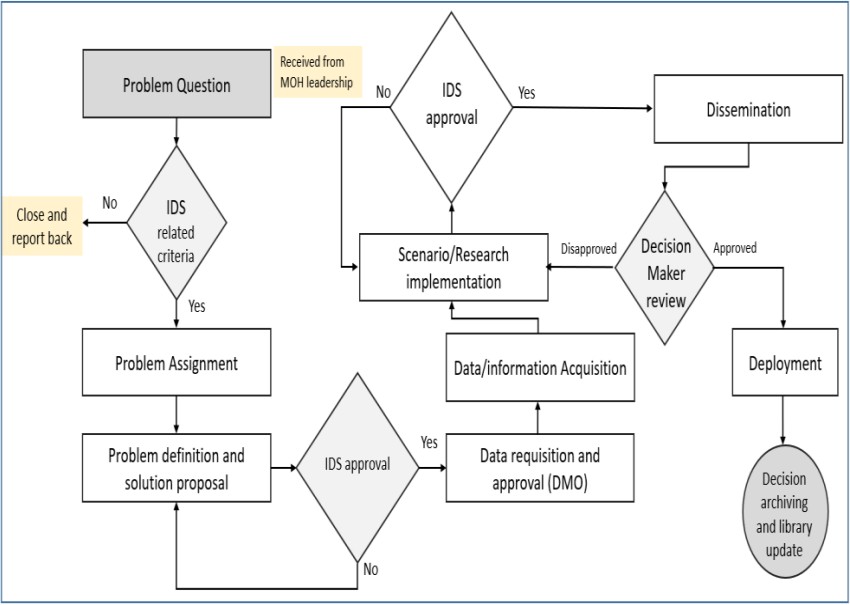

**Figure 3.** Process flow of the Informed Decision Support Framework.

Overall, the IDSF proposed for the health sector appears to be a comprehensive and well-structured approach to informed decision making. By addressing a range of decision types and utilizing a thorough process flow, the framework aims to ensure that the organization is able to make informed and accurate decisions that drive better outcomes.

## 4. Methodology

The data collection tool used also has extensive coverage. The aim of the study was to develop an informed decision support framework for Saudi Arabian organizations. As the organization becomes complex and there is an increase in data available to decision makers, there is an inherent need to make sure that the leaders have sufficient tools to make informed decisions based on the information available to them. Therefore, the research here has adopted a multi-faceted research approach. For this study, we opted for a qualitative research approach [29]. The research has explored into an intensive literature review, which included relevant case studies from across the world, and interviews with different decision makers in the Saudi organization which will be introduced in this section.

### 4.1. Data Collection

The data ware gathered utilizing a qualitative methodology by conducting interviews with decision makers due to the study's experimental character and the limited time available [30,31]. The decision to use the qualitative methodology and interviews to collect data was informed by the elements of the study requirements, and the time limitations relating to data collection. Furthermore, the merits of qualitative research over those of quantitative research were also part of the reasoning for the choice. Quantitative research approaches are designed for the collection of numerical data, which can be applied in the measurement of variables [32]. Usually, the quantitative data are structured and statistical and the results obtained are objective and conclusive. Furthermore, the approach uses the grounded theory, which depends on a systematic analysis of the collected data [32]. Furthermore, the quantitative research approach provides the necessary support when needed to draw general conclusions from the research and predict potential outcomes. To power this study approach for data collections, researchers tend to choose for surveys [32]. These tools are considered flexible, cost-effective, and they allow the collection of data from an extensive sample size.

However, this research is focused on organizations in Saudi Arabia, which required the inclusion of Saudi organizational decision makers. Consequently, the idea was to identify whether the targeted organization had the ideal informed framework to allow informed decision making or what they felt was necessary towards the development of an ideal tool to assist them in the decision making process [33]. Ultimately, there were five participants who were mainly decision makers in the organization. The aim was to obtain responses from all levels of leaders, including both top-level and low-level managers. The decision to select five participants was informed by the findings of Crewell who recommended 5 to 25 participants and Boyd (2001) who recommended 2 to 10 participants provided that the study had thematic redundancy [34].

The interviews focused on two main sections. The first section was an introduction to the study and it involved collecting data about the participant. The second section focused on answering the seven interview questions and obtaining responses to the overall research aim.

### 4.2. Interview Role and Sample

An interview with a few of decision makers in the health sector were conducted to obtain their responses to the questions that were prepared.

The role of the interviewer can be summarized as follows:

1- Getting ready for the interview.
2- Finding respondents and soliciting their cooperation.
3- Addressing any misunderstandings or worries.

4- Watching the level of the answer's clarity.
5- Documenting the answers to start the analysis phase.

## 5. Results and Analysis

Here, the results are presented based on the data collected. They inform the way forward for the development of an informed decision framework to support decision making in the Saudi organization. There were five interviewees who responded to seven semi-structured interview questions. All the five interviewees were leaders in five different departments in the organization. Their personal details and the names of their organizations are left out from the report to ensure confidentiality.

When analyzing the data, the researcher searched for data familiarity, which meant reading the responses and understanding the data offered, observing the impressions, and obtaining all the necessary data from the myriad of information offered [33]. Additionally, through the analysis, the key questions that needed to be answered were identified. Furthermore, there was a focus on answering each of the seven interview questions and comparing the results to the literature findings [35]. Coding and indexing data during the analysis process also proved critical as it enabled us to group the information based on various common elements including ideas, behaviors, concepts, phases, and interactions, among others [36]. Coding also made it possible to manage the information, and obtain the required answers from the bulk of information offered by the interviewees. Therefore, Table 3 summarizes and facilitated the analysis phase. The interviews were had five participants, and the interviewer took between 30 min and an hour to complete with each participant. There were seven questions, and each of the participants had a unique take on the responses given their experience and organizational setup. Different sectors within the organization were represented, comprising representative sample to handle the concept of IDSF within Saudi Arabian organizations.

*Interviews Answers Analysis*

A.    Question One

The first question asked the interviewees, who were decision makers in the organization, about their thoughts regarding the present decision making processes within their organization. It also required them to express whether or not they could guarantee the veracity of the decisions. All the interviewees shared their opinions based on what they had experienced within their organizations. For the most part, all the departments represented had some form of a decision- making process, but the differences emerged when describing how it worked, and how veracity could be guaranteed. According to P1, their organization was hierarchical, and the decision making process required different individuals to participate based on their levels of authority. They also highlighted the importance of the type of decision needed, the urgency, or the overall strategy affected.

> *"As a decision-maker, I am part of the process, and I play my role based on the type of decision being made. While in some cases, and especially when strategy is involved, the decision is largely top-down, there are others where the team at the bottom present options for validation. Therefore, it depends on the type of decision being made, and the reason and urgency involved."*

A similar stance was taken by P2 who also indicated that strategic decisions were made by the top leaders in their organization. P3 stated the following:

> *"Necessary to identify the authority of the individuals who can work on the decision-making process. Finally, the decision-makers have to seek for the relevant information, data and resources that they intend to use during the process."*

Therefore, all the participants understood what the question required of them, and they shared their opinions based on the operations within their sectors. They also highlighted the need for improvements to ensure the validity of the solutions offered during decision making processes.

B.    Question Two

The second question investigated the issues of systems within the organization and what should be used when dealing with decision making. All the respondents had views and opinions regarding potential systems for use, whether they were within their sectors, or whether they understood their importance based on industry understanding. P3 stated that the BIU, data and statistics sources were critical for the organization when making decisions. Data and statistics were also selected by P1 who stated the following:

*"In my opinion data and statistics allow even a perform unfamiliar with an issue to make a conclusive and informed decision."*

P5 highlighted two important elements, which included an internal system within an organization dedicated to support decision making. They also mentioned the importance of center operation from which all relevant health information can be accessed to inform the basis of decision making for an entity.
He said the following:

*"Firstly, an organization's internal system that supports decision making . . . Secondly,* Centers Operation *which has the access to all the relevant health information within the organization."*

C.    Question Three

The third question asked the interviewees about the decision criteria that should be taken into account prior to starting the decision making process. P4 stated that there was a need to understand the type of decision that was required, and in the event that it was a strategic decision for the organization, an informed alternative was necessary. It would be powered by obtaining the necessary data and exploring potential impact. P3 stated the following:

*"The initial process requires defining the problem and identify how it impacts on the organization."*

For most of the responses, the interviewees highlighted the importance of defining the problem to understand the level of authority required to resolve it and allow the organization to find the necessary resources to support the process. P1 also introduced an essential concept by adding that it was necessary to assess the impact that the problem or ultimate decision would have on the organization, which would inform the level of authority that the decision maker required.

*"One needs to identify the actual problem. It means that the problem is defined and each of its relevant elements presented. At this stage, the identified problem is classified in line with its impact within the organization."*

D.    Question Four

The fourth question requested the interviewees to state the sources that had the biggest impact on decisions. While the sources of the ultimate information to make the decision could be diverse, P1 stated the following:

*"There could be multiple sources, but one of the determinants is the actual problem. Once it is analyzed, one can find the most ideal approach to handle it and find a solution."*

One of the sources that they highlighted as important was industry data from which the organization could benchmark what its peers were doing. Internal sources of information, both digital and non-digital, were selected as critical in the decision-making process by P2. Their argument was that most organizations had a lot of data on their historical performance and the results of different actions, which they had not taken into perspective when implementing the selected decisions. P3 had an extensive list of the vital information that their organization required and the sources that they found essential in the process. He said the following:

*"Customer surveys, market research, financials, and the related reports, opinions of experts in the field, such as consultants, lawyers, and financial advisors . . . data from external sources, such as government regulatory agencies."*

P5 selected the organizations within the industry that they found critical in the process, and according to them, all the statistics and data collected through customer surveys, BIU and center operation were vital sources for their organization.

E.　　Question Five

The fifth question asked about the effectiveness of the decision-making process when using digital and non-digital sources. According to the respondents, there was evidence of the fact that they understood how the world was moving away from analogue or non-digital formats toward digital formats. Accordingly, a combination of the two sources was selected as the most effective option by most participants. P4 highlighted the importance of considering to use non-digital sources since, according to them, digital sources were often inaccurate.

*"A business must utilize both digital and non-digital data sources. The digital sources, however, are not always accurate. In order to help the decision-making process, it is crucial to take into account non-digital sources."*

P5 stated that the use of digital sources gave the decision maker access to a wide range of information, and the process of retrieving non-digital materials was time-taking, which could delay the decision making process.
He said the following:

*"When coupled with the swiftness of digital technology and authenticity of non-digital materials, a decision-maker has access to some of the most critical information tools necessary."*

F.　　Question Six

Question six asked about the importance of DMO in the decision decision making process. P3 stated that they were critical as tools of controlling the decision-making process.

*"Limiting access to information, data and potential actions enable organizations control the decision-making process."*

Additionally, they ensured that sensitive data within the organization necessary for the decision-making process was only accessible to authorized parties. P5 stated that DMOs were necessary to ensure that individuals only made the necessary input according to their level of authority in the sense that if a person was only required to save information, they could not retrieve it and edit without getting the proper authority. P4 stated that DMO served two main purposes and said:

*"It's the central unit to guide the decision-making process to the right information, it gives a clear indication of the information's precise source and assure the right access to the data."*

Therefore, the importance of DMO was evidently understood by all the participants, and they highlighted the need for every organization to have such a system within its processes.

G.　　Question Seven

Finally, the last question asked the interviewees to state whether or not they believed that the addition of center operation, BIU, statistics, data, and IDS would improve the decision making process' accuracy. For the most part, all these tools were thought to be important due to their impact on access to data and statistics, which would increase their impact on decision accuracy. According to P3,

*"Data and statistics form the backbone of any decision."*

P4 highlighted the importance of aligning these sources to ensure that the organization prevented duplicity. Finally, P5 stated that the tools improved the process, and they were

well-planned and could be easily audited. They also ensured that the decision-making process presented the required relevance to the organization.

*"An accurate decision-making process is one that is well planned, and easily audited. When multiple tools are added and they interact with ease, it becomes an ideal tool to promote the framework and its operations."*

Finally, Table 4 depicts the interview questions as well as the interviewees' responses to each question.

**Table 4.** Summary of each participant response for all interview questions.

| INTERVIEW ANALYSIS TABLE | | | | | |
|---|---|---|---|---|---|
| Question # | P1 | P2 | P3 | P4 | P5 |
| Q1 As a decision maker, what do you think of the organization's present decision-making process? And how can you guarantee the veracity of the decision? | × | × | × | × | × |
| Q2 Considering that you interact with a variety of systems within the company, what do you think of the systems should be taken into consideration when making decisions? | × | × | × | × | × |
| Q3 What do you think about the decision criteria should be considered before getting started the process of decision-making? | × | × | × | × | × |
| Q4 What are the sources that could have the biggest impact on the decision? | × | × | × | × | × |
| Q5 How effective is the decision-making process when using digital and non-digital sources? | × | × | × | × | × |
| Q6 How crucial is the DMO's presence in the decision-making process? | × | × | × | × | × |
| Q7 Do you believe adding (centers operation, BIU, statistics, data and IDS) will improve the decision-making process' accuracy? | × | × | × | × | × |

## 6. Discussion and Outcomes

The research analyzed the industry problems within Saudi Arabia and found a gap in the decision making processes within organizations. Fundamentally, there was a lack of systems implemented or implementable by organizations at large to allow their decision makers to make decisions from an informed perspective. For this reason, the researcher interrogated the literature and interviewed five decision makers from the health sector regarding the possibility of implementing an informed decision support framework that would improve the processes. The factors for the multicriteria sets were selected based on a comprehensive analysis of the relevant literature and consultation with domain experts. The selection process aimed to identify factors that are commonly considered important in decision making within organizations. These factors were then further refined through iterative discussions and a consensus among the research team.

The initial research questions focused on how the decision makers ensured the accuracy of the decisions they made, and what proper processes were in place to govern and control the results of the decisions. Therefore, the findings analyzed previously were meant to inform whether their proposed framework would be applicable within these organizations, or whether it would improve the accuracy of the existing processes.

One of the issues that became clear from the beginning was that most of the organizations lacked an informed framework within their entities. In some cases, there were no well-planned processes, which guided the decision makers on areas of focus. As the researchers had found out during the literature review, there was a gap within Saudi

organizations in the sense that a model applicable to all organizational decision making processes was lacking. Furthermore, there was no specific model that could be applied by all decision makers, which necessitated the implementation of the proposed model.

When asked about the tools that would be necessary to incorporate into the system, the participants were vocal regarding the importance of tools aligned to data and statistics. These findings aligned to an element that most of the models analyzed during the case study presented. For instance, Liang [27] had a source of evidence as part of their model, and so did Di Matteo [18], who incorporated both external and internal data as part of their database component.

Another element that came up as critical for the framework was the concept of time, which some of the results indicated was critical based on the decision that needed to be made. Essentially, when a decision was needed swiftly, the sources of data that could be used would be mainly digital given their ease of retrieval and analysis. Similar views were held by Almalki [16] as they highlighted the challenges of the implementing information systems. The organizational structure was also a critical issue revealed by the results as it determined how the decisions were made. Even in organizations that lacked clear systems, they still followed the levels of authority, which governed their entity. This is in agreement with the findings of Al Shobaki and Abu-Naser [25] who stated that the levels of authority were critical in the development of effective informed decision support frameworks.

The decision makers interviewed in the process were in support of the elements and tools presented in the proposed framework. They agreed that they would be compatible to their organization, and they would improve the accuracy of the decisions. Furthermore, they would develop a formal process that would guide all the decisions within their sectors.

The paper presented critical elements within the proposed framework, which resolved some of the gaps identified during the literature review. The use of foreign non-digital sources has been largely left out by other models, including what is described in Figure 1, which fail to recognize or mention them.

Contrary to the options presented in these frameworks, the proposed framework has an option to use both digital and non-digital sources, which the interviewees also supported as being a critical source of information for their organizations during the decision making process. Furthermore, several cases were found of some form of decision framework in Saudi Arabia, they were not applicable to all organizations. Therefore, the proposed framework also resolved this problem by presenting a model that can be easily replicated and adapted to any organization. Therefore, through the proposed model, the paper resolved major gap issues identified during the literature review process.

However, the findings supported most of the views that were raised when identifying the research problem. There was a general agreement that Saudi Arabia lacked a general IDS framework that could apply in all organizations. In addition, the researcher concluded that the introduction of the proposed framework would be welcome for the organizations, and it was also necessary as a tool for improving the accuracy of decisions. Additionally, sources of data and statistics came up as essential elements of the framework, and they were considered to strengthen the authority of the decisions.

Therefore, the implementation of the proposed framework for all entities within Saudi Arabia is recommended to assist in their decision making frameworks. It is also recommended that they adjust its processes and elements to fit into their structure and organizational processes. In addition, the automation of the scenarios to be adapted and enhanced through a digitalized DSS framework is recommended. The components have been listed in Table 5.

**Table 5.** Discussion of the components.

| Component | Literature Finding | Research Finding | Similarity | Differences |
|---|---|---|---|---|
| Process and Accuracy | the decision framework and DSS requires data to ensure accuracy and reliability of the processes [11,15,21,22]. Therefore, the quality of data is critical in the process and accuracy. | Data make up a significant part in ensuring the effectiveness of a decision-making system | Data is critical in guaranteeing the effectiveness of the process, and the accuracy of decisions made. | The only differences emerge from the inclusion of specific elements and tools into the system, which guarantee the accuracy and process in addition to data, such as IDS and NHCC |
| Internal & external data | Data is a critical component in the decision-making process [9,13,18,27] | data falls into various categories based on its application in the decision process and format. It can be digital nor non-digital. | the importance of data in the framework is highlighted in both cases. | In the literature review non-digital sources are not expressively defined and explained in relation to their importance. |
| Non-digital sources | The literature review did not capture signficant information on the use of non-digital sources | The findings highlighted the importance of non-digital sources due to their authenticity and availability in organizations. | None | Literature review did not include the component as part of materials discussed. |
| Statistics | Statistics also make up essential information sources in the decision-making process [10]. | Statistics are vital, which necessitates the inclusion of both internal and external sources. | The identification of the importance of statistics in decision-making is evident in both the study and reviewed literature. | The literature does not expound on the importance of including both internal and external statistics from both digital and non-digital sources. |
| Data management and autherity | The literature did not include DMO | DMO is explained as critical in the decision-making process of any organization as it presents the levels of authority and limits access to critical information to authorised personnel only. | None | Current research expounds on the importance of DMO, which is an apparent gap in literature. |
| Decision Criteria | The decision criteria is essential in the proces of decision making as evidenced in the frameworks, such as the MCDA that presents its vitalness in complex problems [11]. | The decision criteria is fundamental in decision making processes of any organization. | The importance of the decision criteria | Greater emphasis and description is shared in the current research than literature. |
| Applicability | The frameworks are only appliacable in their specific fields | The proposed framework can apply in any organaization | all models focus on enhancing decision making | Unlike other frameworks, the proposed model can be used in all organizations in any part of the world. |

## 7. Conclusions

The informed decision support framework has the potential to dramatically enhance the precision and effectiveness of decision making in a wide range of industries and organizations. The implementation of DSS across the Gulf region, particularly Saudi Arabia, has not, however, been without obstacles. In the absence of appropriate frameworks in the region, the effectiveness and impact of these systems, as well as the capacity of businesses to make informed and accurate decisions, have been compromised. It is important to note that the interviews focused on gathering insights into the decision making processes and needs of the stakeholders within Saudi organizations. The purpose was to understand their perspectives and gather real-world inputs to inform the development of the proposed framework. The interviews served as a means to validate the relevance and applicability of the framework in the context of Saudi Arabian organizations. For future studies, what is needed is to conduct a similar study from a survey perspective to increase the scope that it could reach and compare the findings. In addition, future studies are also proposed that focus on how the proposed framework could be applied in each of the sectors within Saudi Arabia. Additionally, a study on how to incorporate the time as a factor that affects the decision making process is also recommended.

**Author Contributions:** Conceptualization, M.A. (Mohammed Alojail) and M.A. (Mohanad Alturki); methodology, S.B.K.; validation, M.A. (Mohammed Alojail) and S.B.K.; formal analysis, M.A. (Mohammed Alojail) and M.A. (Mohanad Alturki); resources, M.A. (Mohammed Alojail); data curation, M.A. (Mohanad Alturki); writing—original draft preparation, M.A. (Mohammed Alojail) and M.A. (Mohanad Alturki); writing—review and editing, S.B.K.; visualization, S.B.K.; supervision, M.A. (Mohammed Alojail) project administration, M.A. (Mohammed Alojail). All authors have read and agreed to the published version of the manuscript.

**Funding:** The authors extend their appreciation to the Deputyship for Research and Innovation, "Ministry of Education" in Saudi Arabia for funding this research (IFKSUOR3-176-2023).

**Conflicts of Interest:** The authors declare no conflict of interest.

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
