# Peer review of "An Informed Decision Support Framework from a Strategic Perspective in the Health Sector"

_information, doi:10.3390/info14070363_

Round 1

Reviewer 1 Report

Dear Author/s,

Many thanks for offering me the privilege to review your paper entitled “An Informed Decision Support Framework From Strategic Perspective in Health Sector”. Despite the originality of topic, I’d recommend some minor changes as follows:

- “Abstract”: I’d advice to reinforce and highlight originality, practical, and theoretical implications in it. Authors should better emphasize the research goals as well as the research design, placing more emphasis on the state of the art and on contributions of the paper.

- “Introduction”: Firstly, starting from the beginning, I’d suggest Authors to better explain the focus of the research and to specify the scope of the paper. Secondly, please try to revamp the Introduction structure as follow: (i) define the contest of the analysis; (ii) clearly explain the gap in the literature that the paper wants to fill; (iii) point out the originality of the article (iv) describe the structure of the paper. Thirdly, I’d like to suggest Authors to better outline the scope of the research since from the Introduction section. Fourthly, certain unclear and long fragment sentences have affected the organization of research idea starting from the first paragraph. Please, revamp the whole section.

- “Related Works”: In this section, it is recommended that Authors give a detailed discussion on each of the frameworks used and the relationship deduced from these frameworks to support this study.

Please, you can consider these international studies in management to enrich your literature review:

·      Orlando, B., Ballestra, L. V., Magni, D., & Ciampi, F. (2021). Open innovation and patenting activity in health care. Journal of Intellectual Capital, 22(2), 384-402.

·      Magni, D., Scuotto, V., Pezzi, A., & Del Giudice, M. (2021). Employees’ acceptance of wearable devices: Towards a predictive model. Technological Forecasting and Social Change, 172, 121022.

- “Research methodology” and “Results”: these sections appear well constructed. Well done!

- “Discussion and Conclusion”: Since I deem that the discussion is relevant to confute or support previous research, I’d reinforce this section properly. Yet, starting from the findings, I would suggest Authors to explain better the novelty of results and the main theoretical but also managerial implications of the paper. Alongside, please strengthen the discussion along with the rest of the article.

Quality of communication

The quality of communication is good. Nonetheless, a professional proof-reading would certainly increase the overall quality of the paper, thus meeting the international standards for peer-reviewed research.

I hope my advice will be useful for a further improvement of your paper.

Best Regards and Good Luck.

Quality of communication

The quality of communication is good. Nonetheless, a professional proof-reading would certainly increase the overall quality of the paper, thus meeting the international standards for peer-reviewed research.

Author Response

MDPI

We are very much thankful to the Editor, Associate Editor and all of the Reviewers for their constructive comments and suggestions. We are really happy that our first updated version of the manuscript with the responses of the reviewers’ comments is accepted. Now responses and changes are made on the paper based on the Editor’s comments are noted below. We marked updates are highlighted with yellow colour in the revised manuscript.

We hope this version is acceptable.

Best wishes

Sincerely Yours,

Surbhi Bhatia Khan

Manuscript ID: information-2388636

Title: An Informed Decision Support Framework from Strategic Perspective in Health Sector

Responses of the comments from the Reviewers

Reviewer 1

Comment 1:  

- “Abstract”: I’d advice to reinforce and highlight originality, practical, and theoretical implications in it. Authors should better emphasize the research goals as well as the research design, placing more emphasis on the state of the art and on contributions of the paper.

Response: Thank you very much for your comment. As per your suggestion, we rewrite the abstarct, highlighted in yellow colour in the revised manuscript. We hope you will recognize our effort.

Comment 2:  “Introduction”: Firstly, starting from the beginning, I’d suggest Authors to better explain the focus of the research and to specify the scope of the paper. Secondly, please try to revamp the Introduction structure as follow: (i) define the contest of the analysis; (ii) clearly explain the gap in the literature that the paper wants to fill; (iii) point out the originality of the article (iv) describe the structure of the paper. Thirdly, I’d like to suggest Authors to better outline the scope of the research since from the Introduction section. Fourthly, certain unclear and long fragment sentences have affected the organization of research idea starting from the first paragraph. Please, revamp the whole section..

Response: Thank you very much for your valuable advice. Changes have been made accordingly, highlighted in yellow colour in the revised manuscript.

Comment 3: “Related Works”: In this section, it is recommended that Authors give a detailed discussion on each of the frameworks used and the relationship deduced from these frameworks to support this study.

Please, you can consider these international studies in management to enrich your literature review:

·      Orlando, B., Ballestra, L. V., Magni, D., & Ciampi, F. (2021). Open innovation and patenting activity in health care. Journal of Intellectual Capital22(2), 384-402.

·      Magni, D., Scuotto, V., Pezzi, A., & Del Giudice, M. (2021). Employees’ acceptance of wearable devices: Towards a predictive model. Technological Forecasting and Social Change172, 121022.

Response: Thank you for your suggestion. Changes have been made accordingly, highlighted in yellow colour in the revised manuscript. The references were found relevant and cited.

Comment 4: “Research methodology” and “Results”: these sections appear well constructed. Well done! 

Response: Thank you for your comment and appreciating our work.

Comment 4: Discussion and Conclusion”: Since I deem that the discussion is relevant to confute or support previous research, I’d reinforce this section properly. Yet, starting from the findings, I would suggest Authors to explain better the novelty of results and the main theoretical but also managerial implications of the paper. Alongside, please strengthen the discussion along with the rest of the article.

Response: Thank you for your comment, the discussions have been added and updated as needed accordingly.

Reviewer 2 Report

Thank you for giving me the opportunity to evaluate this manuscript. The authors did a very good job in collecting and presenting few frameworks for decision making. Unfortunately there are few important missing:

1- how they select the factors for the multicriterial sets? 

2- how they are studied - no connection with the interviews?

3- the figures from 1 to 9 - are subject of copyright?

4- the authors proposed a framework and a process flow and after they have a verification/validation interviews - in my opinion the steps are not in a logic sequence

5- only 5 interviews & 7 questions, in my opinion are to few to be relevant for a multicriterial decision framework. Also the link with the criterion set and the proposed framework is not done.

6- the discussions are consistent but it is hard to put thinks together 

7- the paper as it is do not look like a coherent work, it is more like a puzzle pieces drop in a box

The language is OK from my perspective

Author Response

MDPI

We are very much thankful to the Editor, Associate Editor and all of the Reviewers for their constructive comments and suggestions. We are really happy that our first updated version of the manuscript with the responses of the reviewers’ comments is accepted. Now responses and changes are made on the paper based on the Editor’s comments are noted below. We marked updates are highlighted with yellow colour in the revised manuscript.

We hope this version is acceptable.

Best wishes

Sincerely Yours,

Surbhi Bhatia Khan

Manuscript ID: information-2388636

Title: An Informed Decision Support Framework from Strategic Perspective in Health Sector

Responses of the comments from the Reviewers

Reviewer 2

Comment 1:  

- how they select the factors for the multicriterial sets? 

Response: Thank you very much for your comment. They have been selected based on a comprehensive analysis of the DSS Frameworks in the literature and the same has been highlighted in the text.

Comment 2:  how they are studied - no connection with the interviews?

Response: Thank you very much for your valuable advice. The same has been highlighted in the text. 7 questions were asked from the stakeholders based on the research gaps identified in the literature and the researchers were interviewed sequentially in a systematic manner to complete the qualitative analysis.

Comment 3: the figures from 1 to 9 - are subject of copyright? 

Response: Thank you for your suggestion. Some figures have been deleted and some are redrawn to avoid any copyright issue.

Comment 4: - the authors proposed a framework and a process flow and after they have a verification/validation interviews - in my opinion the steps are not in a logic sequence

Response: Thank you for your comment, the steps have been explained for clear explanation.

Comment 4: only 5 interviews & 7 questions, in my opinion are to few to be relevant for a multicriterial decision framework. Also the link with the criterion set and the proposed framework is not done.

Response: Thank you for your comment, we intend to do the analysis further with the criterion set as a future direction.

Comment 5: the discussions are consistent but it is hard to put things together 

Response: Thank you for your comment, the discussions have been reconstructed again for better clarity.

Round 2

Reviewer 1 Report

Accept

Reviewer 2 Report

I agree with the actual version of the paper.

The language is OK in my opinion.